# SimGroupAttn: Similarity-Guided Group Attention for Vision Transformer to Incorporate Population Information in Plant Disease Detection

WangYang Wu[*1,2], Ribana Roscher[2], and Niklas Tötsch[1]

[1]Crop Science, Bayer AG, Germany
[2]University of Bonn, Germany
{wangyang.wu, niklas.toetsch}@bayer.com
ribana.roscher@uni-bonn.de

## Abstract

In this paper, we address the problem of Vision Transformer (ViT) models being limited to intra-image attention, which prevents them from leveraging cross-sample information. This is highly relevant in agricultural data such as plant disease detection, an important challenge in agriculture where early and reliable diagnosis helps protect yields and food security. Yet existing methods often fail to capture subtle or overlapping symptoms that only become evident when considered in a population context. Our approach *SimGroupAttn* extends masked image modeling by enabling image patches to attend not only within their own image but also to similar regions across other images in the same batch. Guided by cosine similarity score which is trained jointly with model weights,*SimGroupAttn* incorporates population-level context into the learned representations, making them more robust and discriminative. Extensive experiments on PlantPathology dataset demonstrate that our approach outperforms Simple Masked Image Modeling (SimMIM) and Masked Autoencoders (MAE) in linear probing and classification task. It improves top-1 accuracy by up to 6.5% in linear probing for complex classes and 3.5% in classification compared with the best baseline model performance under the same settings.

## 1 Introduction

In recent years, Vision Transformers (ViTs) [1] have attracted considerable interest for their ability to model long-range dependencies across image patches through self-attention. This global attention mechanism enables ViTs to capture richer visual context, making them particularly effective for complex visual recognition tasks. However, most ViT architectures focus attention solely within a single image, while this design is effective for many scenarios, it limits the ability to incorporate broader context. In fields like agriculture and biology, samples frequently share phenotypic traits or disease patterns that are only meaningful when viewed in the context of other data points.

Existing masked image modeling methods like Masked Autoencoders (MAE) [2] and SimMIM [3] have shown strong performance by reconstructing masked patches and encouraging robust representation learning. Yet, because they operate on isolated images, they struggle to capture the broader patterns and variability that are essential for robust representation learning in population-sensitive domains.

To address this limitation, we introduce *Similarity-Guided Group Attention (SimGroupAttn)*, a mechanism that extends self-attention beyond individual images. Instead of restricting attention to intra-image patches, it allows each patch to attend to similar regions across other images within the same batch. These cross-image links are guided by similarity scores, allowing the model to learn from population-level information during training. We integrate SimGroupAttn into a masked image modeling (MIM) framework [4–8], where the added context improves reconstruction of masked patches and strengthens representation learning.

In summary, this paper makes three key contributions:

1. We propose *SimGroupAttn*, a novel attention mechanism that incorporates population-level information into ViTs by enabling similarity guided cross-image attention.

2. *SimGroupAttn* jointly learns the similarity scoring and model parameters, allowing the similarity function to adapt during training. This leads to more effective task-specific modeling and improved generalization in diverse samples.

3. We show that *SimGroupAttn* not only outperforms existing MIM-based frameworks in representation learning, but also significantly improves downstream classification tasks involving classes with highly overlapping features.

In summary, our results highlight the value of modeling cross-sample relationships, especially in domains where complex feature patterns are key to accurate visual understanding.

---

*Corresponding Author.

Proceedings of the 7th Northern Lights Deep Learning Conference (NLDL), PMLR 307, 2026.

## 2 Related Work

**Masked Image Modeling (MIM).** [4–8] has become a popular self-supervised learning strategy for ViTs. The key idea is to randomly mask image patches and train the model to reconstruct the missing content, thereby encouraging the encoder to capture meaningful visual representations. Masked Autoencoders (MAE) [2] and SimMIM [3] pioneered this approach for ViTs. MAE adopts an asymmetric design, masking most patches (typically 75%) and encoding only the visible ones, while a lightweight decoder reconstructs the masked content. This reduces computation and allows the encoder to focus on semantic abstraction. SimMIM offers a simpler alternative by reconstructing masked patches directly with the encoder, yet still achieves competitive results, showing that effective reconstruction-based pretraining can be realized with minimal architectural changes. While both MAE and SimMIM have demonstrated strong performance in downstream recognition tasks, they share a common limitation: attention remains confined to intra-image patches. This makes them less effective in domains where cross-sample relationships are crucial.

**Plant Disease Classification.** Deep learning has been widely applied to plant disease classification, with models such as AlexNet and ResNet demonstrating strong early results [9–11]. More recently, ViTs [1] have gained attention in this area, with several works exploring ViTs alone or in hybrid combinations with CNNs [12–16]. However, most approaches still focus on individual images and rarely incorporate population-level information, which is particularly important in domains such as agriculture and biology. For instance, a single leaf may exhibit symptoms of multiple diseases simultaneously, making its representation more challenging. A recent study by Eu-Tteum Baek [17] attempted to address this by combining multiple independent ViTs for population-aware classification. While this approach achieves high accuracy, each ViT still operates primarily at the instance level since their training does not involve population-level attention.

**Joint Training of ViTs.** Unifying the strengths of self-supervised and supervised learning for training ViT especially under circumstances of limited data has gained substantial attention. Qian et al. [18] proposed a novel joint training framework where a Vision Transformer is optimized simultaneously using MAE reconstruction loss and a classification loss from the very beginning of training. Unlike conventional two-stage pipelines that separate self-supervised pretraining from downstream fine-tuning, their method integrates both objectives into a unified training loop, allowing the encoder to learn semantically rich and task-aligned representations in a data-efficient manner. This approach is particularly effective in low-data circumstances, as the reconstruction loss acts as a strong inductive bias that regularizes the learning process and helps prevent overfitting. Zhou et al. introduced UniMAE [19], a unified masked autoencoder that supports multi-task training by jointly optimizing for image reconstruction and multiple downstream objectives, including classification and detection. In this study, we also adopt joint training for one of our experiments to address the issue of limited data.

## 3 Method

### 3.1 SimGroupAttn Mechanism

Standard ViT restricts attention to patches within a single image, our *SimGroupAttn* extends this mechanism to enable cross-image interactions. A few key differences are illustrated here:

**Intra-image attention & Cross-image Attention.** Rather than limiting attention to only intra-image patches, each query patch in the input batch attends to its top-$M$ most similar patches within the same image (intra-image) and top-$N$ most similar patches across the batch (cross-image) — resulting in a total of $(M + N)$ attended patches. These connections form an attention graph that guides the attention layer in ViT to cross all patches in a batch. This graph, along with the randomly masked patch embeddings, is then fed into the attention blocks. During attention computation, selected entries in the key matrix $K$ and the value matrix $V$ are indexed according to the attention graph while query matrix $Q$ is computed identically to standard attention. This produces a sparsely refined attention pattern that focuses the computation on the most relevant patches. In standard self-attention, the output is computed as:

$$\text{Attention}(Q, K, V) = \text{softmax}\left(\frac{QK^\top}{\sqrt{D}}\right)V \quad (1)$$

In *SimGroupAttn*, attention is restricted to a similarity-guided neighborhood $\mathcal{N}_q$ for each query patch $z_q$. Let $Q_q = z_q W_Q$, $K_k = z_k W_K$, and $V_k = z_k W_V$ for all $k \in \mathcal{N}_q$ where $W$ are learnable weight matrices that linearly project the inputs into query, key, and value matrix and $D$ is the feature dimension. The attention is computed as:

$$\text{Attention}(z_q) = \sum_{k \in \mathcal{N}_q} \alpha_{qk} V_k \quad (2)$$

where the attention weights $\alpha_{qk}$ are given by:

$$\alpha_{qk} = \frac{\exp\left(\frac{Q_q K_k^\top}{\sqrt{D}}\right)}{\sum_{j \in \mathcal{N}_q} \exp\left(\frac{Q_q K_j^\top}{\sqrt{D}}\right)} \quad (3)$$

**Similarity Guidance.** A patch-pairwise cosine similarity scoring system measures how similar two patches are and then helps to construct the attention graph. For each input batch, pairwise similarity is calculated for all patches in the same batch which forms a similarity matrix capturing relations between patches both in the same image and across different images. This similarity matrix is computed after patches are passed through the patch embedding layer. The motivation of a post-patch embedding similarity is that the model can jointly learn similarity scoring and weights simultaneously. This allows the model to better adapt to various domains by enabling it to combine patch-wise relation learning and weight optimization in one go. In addition, we also set the similarity scores between the same patch to be $-\infty$ when $M$ is smaller than the total number of patches in an image to prevent patches attending to themselves. This aims to maximize population-level information utilization and limit the original signal flowing into reconstruction .

The general workflow of *SimGroupAttn* is illustrated in Figure 1.

## 3.2 Model Architecture

We benchmark our method against two Vision Transformer–based masked image modeling approaches: SimMIM [3] and MAE [2]. Given the relatively small size of our datasets, we scale down the Vision Transformer encoder to ViT-Tiny (ViT-T) configuration. For MAE, we also reduce the decoder's capacity to ensure a comparable number of parameters across models. However, because MAE's performance is highly dependent on decoder design, we retain a relatively larger decoder for MAE.

Table 1 summarises the architectural configurations of the compared methods. All three models share the same encoder design, consisting of a single convolutional patch embedding layer, learnable positional embeddings, and a 12-layer transformer encoder with 6 attention heads of dimension 64, resulting in an embedding dimension of 384 and approximately 23.7M parameters. The main difference lies in the decoder design. MAE employs a deeper attention-based decoder with two layers and three attention heads, resulting in approximate 2.6M parameters. In contrast, both SimMIM and SimGroupAttn adopt a lightweight CNN decoder followed by PixelShuffle, with only two layers and 1.18M parameters.

## 4 Experiments

In this section, we evaluate the effectiveness of our approach from three main aspects: reconstruction quality, representation learning quality and classification performance. For the first two tasks, all models were trained on the PlantVillage dataset [20] using self-supervised Masked Image Modeling (MIM) [2, 3]. We then assess reconstruction quality by visually comparing reconstructed images of different methods. We evaluate representational learning quality through linear probing on a subset of the Plant Pathology dataset containing the most complex classes. To evaluate classification performance, we train all models on the full PlantPathology dataset [21] using a joint method of supervised and self-supervised [18]. In addition, we conducted several ablation studies in the classification task to explore some important factors.

Table 2 shows an overview of our experiment design.

## 4.1 Dataset

We use two different datasets in this study: PlantVilliage [20] and PlantPathology [21].

**PlantVillage dataset.** This dataset contains a large and diverse collection of plant leaf images spanning 38 disease classes across 14 crop species. Original resolution is $256 \times 256$. To ensure consistency and improve the overall quality of the input data, all images were segmented using SAM 2 [22].

**PlantPathology dataset.** It is a publicly available benchmark for image-based plant disease classification that consists of high-resolution images. We used the version with a downsized resolution of $640 \times 427$. Images are annotated with multiple disease categories, including complex cases where leaves exhibit symptoms of more than one disease. Such complex classes demonstrate a more entangled feature space, making linear separation significantly harder.

All images are resized to $224 \times 224$ before feeding into the models. The statistics for each dataset, including the number of images, class counts, and splits used in our experiments, are shown in Table 3.

**Batch Construction.** For SimMIM and MAE, batches are typically randomly sampled from the entire dataset. In contrast, SimGroupAttn requires more careful batch construction. Because attention is applied across the whole batch, it is preferable to group images of the same plant species within a batch rather than sample randomly across species, in order to avoid cross-species attention. Therefore, when training on the PlantVillage dataset, each batch was randomly sampled from images of the same plant type, whereas for the Plant Pathology dataset, standard random sampling was used since all images come from the same plant species.

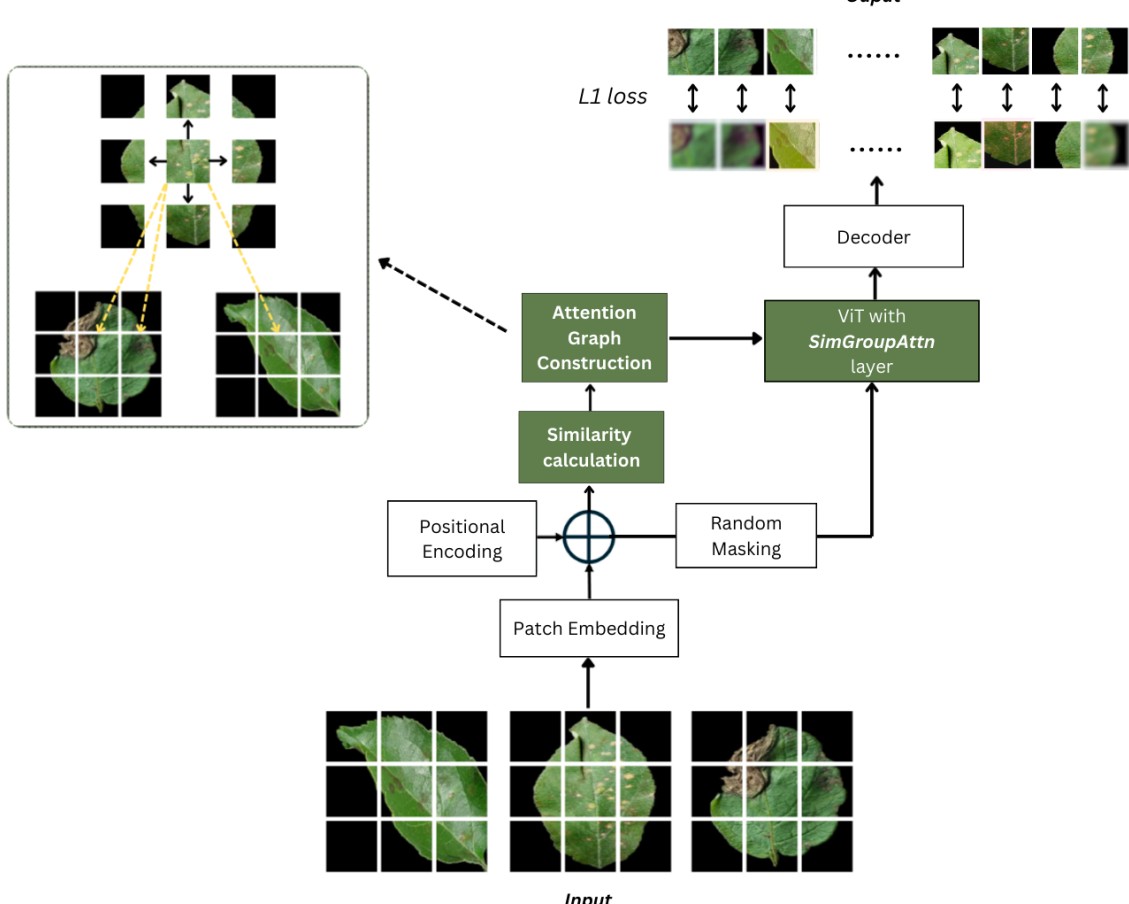

**Figure 1.** An illustration of masked image modeling framework using our *SimGroupAttn* Vision Transformer. Input (bottom) first goes through a patch embedding. Before the randomly masking patches are then fed into the attention block, pair-wise similarity is computed between all patches in the batch to construct an attention graph. Encoded patches are then randomly masked with a mask ratio of 0.5 and fed into the attention block along with the attention graph. The output (top right) of ViT encoder is decoded back to the original image and $L1$ loss is calculated only on the masked patches.

## 4.2 Reconstruction & Representation Learning Performance

### 4.2.1 Training

We trained all three models on the PlantVillage dataset using Masked Image Modeling (MIM). During training, a random mask ratio of 0.5 was applied to all models. Although the original SimMIM and MAE studies adopt higher mask ratios to improve performance, the limited size of our dataset raised concerns that overly aggressive masking could slow down convergence. For this reason, we chose a lower mask ratio. During testing, we used a higher mask ratio of 0.7 to evaluate reconstruction quality and no masking for representation learning since models need to see all image patches to produce a meaningful representation.

To reduce computational cost, particularly for *SimGroupAttn*, we used a patch size of 32, resulting in a total of 49 ($7 \times 7$) patches. In addition, we set both intra-image patches ($M$) and cross-image patches ($N$) to 15. This configuration was motivated by the nature of PlantVillage dataset, where segmented objects are typically centered, and for a majority of images, the object area can be covered by approximately 30 patches. After masking half of them, 15 patches remain in the same image that carry meaningful information. To maintain consistency, the number of cross-image patches was also fixed at 15 so that all models see a similar number of meaningful paths. By fixing on a smaller number of attention connections, we also reduced computational complexity since *SimGroupAttn* requires more memory in attention indexing.

All models were trained on a cluster equipped with 8 NVIDIA A10G GPUs (24 GB each) using identical experimental settings to ensure fair comparison. Pretraining was conducted for 600 epochs with an effective batch size of 32 per GPU. Gradient accumulation was applied every four steps to achieve

**Table 1.** Configuration of encoders and decoders for different methods.

| Method | Encoder | | | | | | | Decoder | | | |
|---|---|---|---|---|---|---|---|---|---|---|---|
| | Positional Embedding | Patch Size | # Depth | # Heads | Head Dim | Dim | # Param | Network | # Layers | # Heads | # Param |
| SimMIM | learnable | 32 | 12 | 6 | 64 | 384 | 23.7M | Conv + PixelShuffle | 2 | - | 1.18M |
| MAE | learnable | 32 | 12 | 6 | 64 | 384 | 23.7M | Attn + Linear | 2 | 3 | 2.6M |
| SimGroupAttn | learnable | 32 | 12 | 6 | 64 | 384 | 23.7M | Conv + PixelShuffle | 2 | - | 1.18M |

**Table 2.** Overview of experiment design.

| Task | Dataset | Training Method | Evaluation Method | Ablation Studies |
|---|---|---|---|---|
| Reconstruction Quality | PlantVillage [20] | Self-Supervised | Visual Inspection | ✗ |
| Representation Learning | PlantVillage [20] & PP [21] | Self-Supervised | Linear Probing | ✗ |
| Classification Performance | PlantPathology [21] | Supervised & Self-Supervised (Joint) | Multiclass Classification | ✓ |

**Table 3.** Overview of datasets used in this study

| Dataset | Image Size | # Classes | # Train | # Test |
|---|---|---|---|---|
| PlantVillage [20] | 256×256 | 38 | 45312 | - - |
| PlantPathology [21] | 640×427 | 12 | 16696 | 1936 |

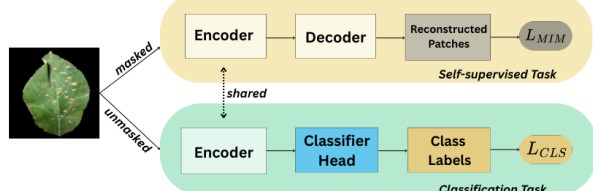

**Figure 2.** Illustration of joint training framework. Input (left) goes through two passes. One computes pixel-wise reconstruction loss for masked image modeling task (top); another one computes classification entropy loss (bottom). Encoder is shared and total loss is summed with a weight $\lambda = 0.1$ for $L_{MIM}$

an effective batch size of 512. The base learning rate was set to 0.0005 with a 50-epoch warm-up phase. AdamW optimization was used with a weight decay of 0.05, and a cosine learning rate scheduler. For the reconstruction loss, SimMIM and SimGroupAttn used L1 loss, while MAE relied on L2 loss.

### 4.2.2 Evaluation

**Reconstruction Performance.** To assess reconstruction quality, we visually inspect the reconstructed images of *SimGroupAttn* and compute the $L1$ loss compared to the baselines. For visual comparison, we emphasize mostly on images of leaves that are affected by one or more diseases or that have irregular shapes, color, patterns, etc. We believe it is relatively harder to reconstruct and requires a richer context in such cases to have a decent reconstruction.

**Representation Learning Performance.** we conduct linear probing on the pre-trained models using PlantPathology dataset. It contains five simple classes and seven complex classes where each leaf is infected by multiple diseases. This poses more challenges since feature space of complex classes is highly entangled and they cannot be easily separated in a linear way. Such cases require models to learn a better context from a population perspective.

### 4.3 Classification Performance

#### 4.3.1 Training.

We adopted a joint training framework for this task, similar to the work [18]. The motivation of adopting such a method is to address the issue of limited data which results in the inefficient learning of inductive bias for ViT. Moreover, the model can benefit

from being able to learn semantically rich and task-specific feature space simultaneously. Each input batch undergoes two forward passes: one through a standard MIM pipeline with a decoder to reconstruct masked patches, where the pixel-wise L1 loss is computed ($L_{\text{MIM}}$); and another through a linear classification head, where the cross-entropy loss ($L_{\text{CLS}}$) is computed. The total loss is defined as

$$L = \lambda L_{\text{MIM}} + (1 - \lambda) L_{\text{CLS}}. \tag{4}$$

We choose $\lambda = 0.1$ to have the model focus more on task-specific learning. Figure 2 illustrates the workflow of the training pipeline. The encoder is shared during training; after training, the decoder is discarded and only the classifier head remains for the classification task. All models were trained on a 4-GPU cluster with identical hardware configurations for 500 epochs, using a base learning rate of 0.0005. We employed 20 warm-up epochs and optimized with cross-entropy loss, incorporating a label smoothing factor of 0.01 to improve generalization and mitigate overfitting.

#### 4.3.2 Evaluation

We perform a series of ablation studies to assess model performance across different configurations, evaluating the top-1 accuracy over the entire test dataset as well as for each individual class to provide a more granular analysis of the model's performance.

To better understand how specific design choices influence the performance of *SimGroupAttn* in this task, we examine several main factors: the number of intra-image patches $\mathbf{M}$, the number of cross-image patches $\mathbf{N}$, and the batch size $\mathbf{B}$.

**Effect of M and N.** $\mathbf{M}$ controls how much information to use from the same image while $\mathbf{N}$ controls how many patches from other images a certain patch can attent to which calibrate the amount of external context to utilize. The balance of $\mathbf{M}$ and $\mathbf{N}$ can have an affect in the model's performance since both Intra and Cross-image information is important in certain scenarios. With a larger $\mathbf{N}$, the model relies more on patterns shared across the dataset, whereas a larger $\mathbf{M}$ keeps the model focused mainly on the context within its own image. To ensure a fair comparison, we maintain the same total number of patches a patch can attend to for all three models, namely, $\mathbf{M + N = 49}$. For the two baseline models, $\mathbf{M = 49}$ and $\mathbf{N = 0}$ in all cases.

**Effect of B.** The batch size $\mathbf{B}$ adjusts the overall diversity of that context by increasing or decreasing the number of images in a single batch. For SimMIM and MAE, this might be a less crucial factor since they do not rely on cross-image attention. However, for *SimGroupAttn*, larger batches are likely to provide a richer population-level context, allowing the model to compare patches in a broader set of samples, whereas smaller batches limit the diversity of reference patches and reduce the benefit of cross-image attention.

# 5 Results

## 5.1 Reconstruction

Since mask ratio during evaluation is high at 0.7, the reconstruction task is substantially more challenging than training since models can only see a minority of patches. Under this setting, large portions of the leaf are masked out, requiring the model to infer missing content from very limited contextual information. In reality, it is common that disease symptoms appear only in small regions of the leaf; if these areas are masked, reconstructing them becomes highly challenging since the remaining visible patches often contain semantically different content.

As illustrated in Figure 3, SimMIM and MAE fail to recover small defective regions, producing reconstructions that resemble generic approximations rather than the original image. Our approach leverages similarities across patches from other images within the same batch, allowing it to recover missing regions more effectively. The reconstructions produced by our model remain closer to the original images and better preserve essential phenotypical

| Method | Single class | | Complex class | |
|---|---|---|---|---|
| | Top-1 (%) | Top-3 (%) | Top-1 (%) | Top-3 (%) |
| SimMIM | **56.9** | **90.5** | 37.8 | 82.0 |
| MAE | 56.5 | 90.2 | 37.5 | 78.9 |
| SimGroupAttn (Ours) | 56.4 | 90.4 | **44.3** | **82.5** |

**Table 4.** Comparison of Top-1 and Top-3 accuracy in linear probing on PlantPathology for single vs. complex class .

features such as shape, texture, and color patterns. This capacity to recover fine-grained details shows the advantage of incorporating cross-image information. Per-class reconstruction errors are provided in Appendix B.1, where **SimGroupAttn** consistently achieves lower $L1$ losses across most species and disease types, demonstrating its superior reconstruction fidelity.

## 5.2 Representation Learning

We evaluate how well models learn representations by conducting linear probing on PlantPathology dataset [21]. Due to the classes are quite imbalanced between simple classes and complex classes, we run linear probing on these two groups separately. In simple class, leaves are only infected by one disease while in complex classes, they are infected by multiple diseases which makes feature space heavily entangled and difficult to linearly separated. As shown in Table 4, in simple classes, performances of all three models are very comparable with a maximum of 0.5% difference in top-1 accuracy and 0.3% in top-3 accuracy. Such differences are likely not to be significant due to randomness such as batch formation. On contrary, in complex classes, *SimGroupAttn* outperforms both SimMIM and MAE, achieving a Top-1 accuracy of 44.3% and 82.5 % Top-3 accuracy, a notable improvement of nearly 6.5% and 6.8% over SimMIM (37.8%) and MAE (37.5%) respectively on Top-1 accuracy. This suggests that although it shows no advantages on simple classes, incorporating population context during pretraining helps achieve better representation learning in cases where data space is highly entangled.

## 5.3 Classification & Ablation Studies

### 5.3.1 Overall Accuracy

As shown in the *Overall* column of Table 5 and Figure 4, *SimGroupAttn* outperforms both baseline models in most configurations. *SimMIM* maintains stable accuracy across batch sizes, whereas *MAE* achieves a noticeably higher accuracy of 59.6% at batch size 32. We suspect this may be due to more optimal learning for a limited number of classes during training. In fact, Table 5 shows that at batch size 32, *MAE* performs exceptionally well on the *PM* class, with a 24.7% increase compared

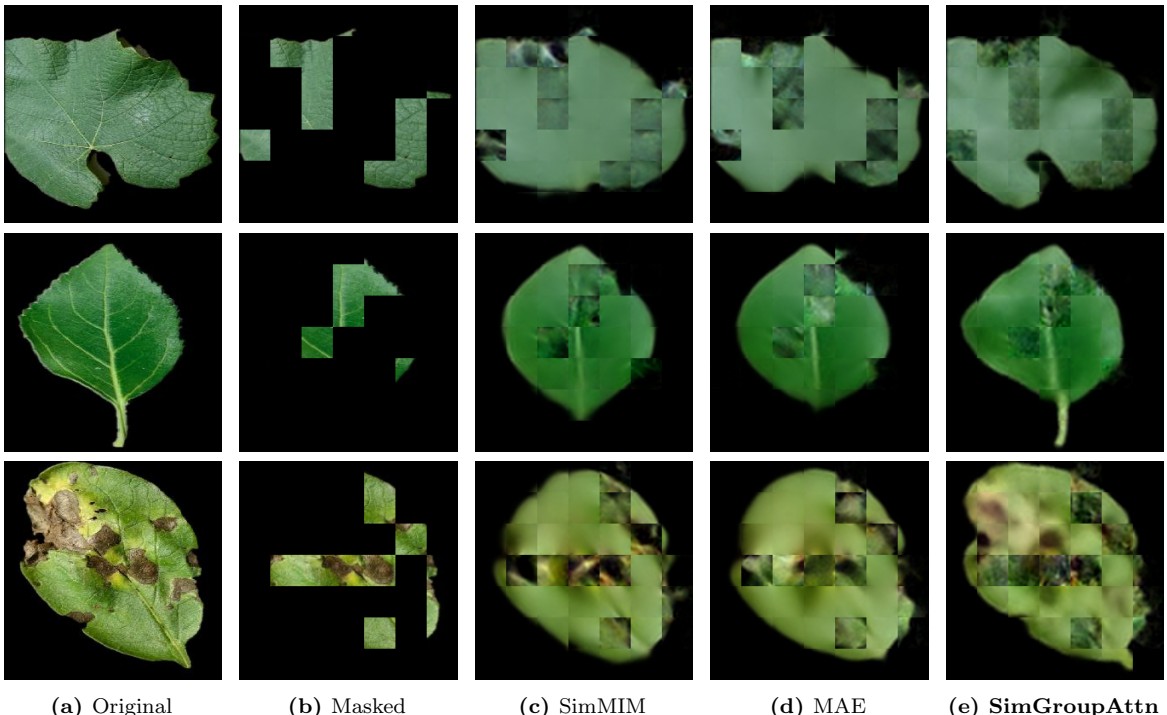

(a) Original     (b) Masked     (c) SimMIM     (d) MAE     (e) **SimGroupAttn**

**Figure 3.** Comparison of original, masked, and reconstructed images for four different examples. Each column corresponds to one method, and each row shows a different input case.

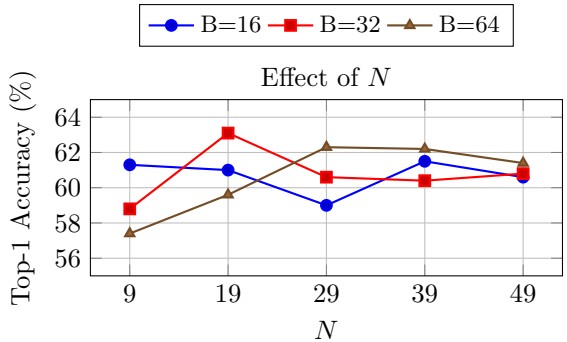

**Figure 4.** Top-1 accuracy of *SimGroupAttn* across different $N$ values, for batch sizes 16, 32, and 64.

to batch size 16. This behavior is unusual, since neither *SimMIM* nor *SimGroupAttn* exhibits such large variation in this class. While *SimMIM* and *MAE* reach their highest accuracies at 56.9% and 59.6% respectively, *SimGroupAttn* achieves a peak accuracy of **63.1%** with $M = 30$, $N = 19$ at batch size 32, representing a 6.2% and 3.5% improvement over the baselines. Even under less optimal configurations, *SimGroupAttn* consistently maintains accuracy above 60%. These results suggest that incorporating population-level information helps the model better distinguish between groups, as cross-image attention encourages the formation of more informative latent representations that improve classification accuracy.

The following paragraphs provide a more detailed breakdown on the effect of $M$, $N$, and batch size $B$ on classification performance.

**Batch Size 16.** As shown by the blue line in Figure 4, at a lower batch size of 16, the performance of *SimGroupAttn* remains relatively stable across different values of $N$, with only a noticeable dip at $N = 29$. This suggests that if the batch size is limited, the model does not benefit from increasing the proportion of cross-image patches because they are statistically unlikely to be more informative than intra-image patches.

**Batch Size 32.** The single best performance of Top-1 accuracy of 63.1% is observed at the configuration: $B = 32$, $M = 30$ ,and $N = 19$ (red line in Figure 4). This configuration substantially outperforms both SimMIM (56.9%) and MAE (59.6%). As $M$ decreases (and $N$ increases), the accuracy gradually decreases to 60.6% in $M = 20$ and 60. 4% in $M = 10$, indicating diminishing returns when too much emphasis is placed on cross-image interactions.

**Batch Size 64.** As illustrated by the brown line in Figure 4, at a higher batch size of 64, performance improves progressively with increasing $N$, peaking at $N = 29$. This trend indicates that when using larger batches, the model is likely to benefit from distributing more attention toward population-level features.

| Method / #Diseases per leaf / #Images | B | M | N | Healthy 1 462 | Scab 1 482 | Rust 1 186 | PM 1 118 | FL 1 318 | SFL 2 68 | RFL 2 12 | SFLComplex ≥3 55 | FLComplex ≥3 27 | PMComplex ≥3 23 | RustComplex ≥3 25 | Complex ≥3 160 | Overall |
|---|---|---|---|---|---|---|---|---|---|---|---|---|---|---|---|---|
| SimMIM | 16 | 49 | 0 | 75.7 | 63.6 | 64.3 | 78.1 | 60.0 | 6.7 | 0 | 0 | 0 | 0 | 0 | 20.0 | 56.7 |
| | 32 | 49 | 0 | 69.8 | 61.8 | 58.5 | 80.0 | 55.7 | 7.1 | 0 | **10.0** | 0 | 0 | 16.7 | 50.0 | 56.9 |
| | 64 | 49 | 0 | 68.4 | 63.3 | 64.4 | **87.5** | 51.5 | 7.2 | 0 | 0 | 0 | 0 | 0 | 40.1 | 56.4 |
| MAE | 16 | 49 | 0 | 72.7 | 63.6 | 62.0 | 58.6 | 51.2 | 0 | 0 | 0 | 0 | 0 | **33.3** | 47.5 | 56.3 |
| | 32 | 49 | 0 | 70.7 | 67.2 | 65.3 | 83.3 | 62.0 | 10.0 | 0 | 0 | 16.7 | 0 | 20.0 | 36.5 | 59.6 |
| | 64 | 49 | 0 | 80.2 | 64.8 | 64.1 | 59.3 | 40.8 | 9.5 | 0 | 7.1 | 0 | 0 | 0 | 40.0 | 55.4 |
| SimGroupAttn | 16 | 40 | 9 | 83.1 | 64.8 | 62.5 | 75.7 | 58.9 | **25.0** | 0 | 0 | 16.7 | 0 | 0 | 45.0 | 61.3 |
| | 16 | 30 | 19 | 78.0 | **73.7** | 57.1 | 78.6 | 59.7 | **25.0** | 0 | 0 | **50.0** | 10.0 | 20.0 | 35.6 | 61.0 |
| | 16 | 20 | 29 | 77.7 | 65.5 | 65.3 | 65.5 | 61.4 | 6.7 | 0 | 0 | 0 | 0 | 0 | 37.8 | 59.0 |
| | 16 | 10 | 39 | 81.3 | 72.7 | 70.5 | 75.0 | 54.8 | 5.9 | 0 | 0 | 0 | 0 | **33.3** | 43.8 | 61.5 |
| | 16 | 0 | 49 | 78.8 | 64.8 | 57.5 | 82.1 | 59.7 | 13.3 | 0 | 0 | 0 | 0 | 0 | 51.1 | 60.6 |
| | 32 | 40 | 9 | 77.7 | 66.4 | 62.9 | 84.0 | 51.2 | 0 | 0 | 9.1 | 0 | 0 | 0 | 53.7 | 58.8 |
| | 32 | 30 | 19 | 82.4 | 67.7 | 70.4 | 67.9 | **64.0** | 6.7 | 0 | 0 | 0 | 0 | 0 | 44.7 | **63.1** |
| | 32 | 20 | 29 | 85.0 | 61.6 | 60.8 | 75.9 | 61.3 | 12.5 | 0 | 0 | 25 | 0 | 0 | 38.9 | 60.6 |
| | 32 | 10 | 39 | 71.1 | 63.2 | **72.9** | 78.9 | 59.3 | 18.8 | 0 | 0 | 0 | 0 | 0 | 45.2 | 60.4 |
| | 32 | 0 | 49 | 79.8 | 66.7 | 53.5 | 77.4 | 61.0 | 5.9 | 0 | 0 | 16.7 | **20.0** | 0 | 45.7 | 60.8 |
| | 64 | 40 | 9 | 75.0 | 63.9 | 62.5 | 85.0 | 55.6 | 0 | 0 | 9.1 | 0 | 0 | 20.0 | 32.3 | 57.4 |
| | 64 | 30 | 19 | 70.0 | 70.5 | 63.6 | 73.1 | 61.3 | 5.9 | 0 | 0 | 28.6 | 0 | 20.0 | 25.0 | 59.6 |
| | 64 | 20 | 29 | 81.8 | 66.4 | 66.7 | 76.7 | 52.9 | 0 | 0 | 0 | 0 | 0 | 0 | **54.5** | 62.3 |
| | 64 | 10 | 39 | **88.8** | 64.6 | 69.1 | 80.0 | 50.0 | 10.5 | 0 | 0 | 0 | 0 | 20.0 | 53.1 | 62.2 |
| | 64 | 0 | 49 | 82.1 | 66.4 | 65.1 | 76.2 | 60.3 | 20.0 | 0 | 0 | 0 | 0 | 0 | 37.9 | 61.4 |

**Table 5.** Top-1 accuracy for different configurations. Number of diseases shown on each leaf in each class is also shown in the header line. *PM, FL, SFL, RFL* are *powdery mildew, frog eye leaf spot, scab frog eye leaf spot*, and *rust frog eye leaf spot* respectively. *Complex* suffix means leaves in a class are infected by multiple disease while having a main disease (prefix). Number of images in test set for each class is also shown in row *#Images*.

### 5.3.2 Per Class Accuracy

In addition to its superior overall performance, *SimGroupAttn* also demonstrates improved per-class accuracy in ten out of twelve classes considering all configurations as shown in Table 5, although there is no single configuration that can outperform both baselines in all classes simultaneously.

For simple classes, *SimGroupAttn* shows a large margin over baselines in particular cases under certain configurations. For example, in class *Healthy*, *SimGroupAttn* shows a improvement of 13.1% and 8.6% compared to the best accuracy in SimMIM and MAE respectively; in class *Scab*, an improvement of 10.4% and 6.5%; also in class *Rust*, 8.3% and 7.6% respectively; while in class *PM*, it falls behind SimMIM and in class *FL*, only a small margin over both baselines is achieved.

In more challenging classes where multiple diseases co-occur on the same leaf, baseline models remain competitive in some cases. For example, SimMIM achieves the best result in the *SFLComplex* class with an accuracy of 10%, and MAE shows comparable performance in the *RustComplex* class with an accuracy of 33.3%. For other classes, *SimGroupAttn* delivers stronger results. For instance, in class *FLComplex*, best performance from our model gives 50% exceeding best baseline accuracy over 33.3%; in class *PMComplex*, *SimGroupAttn* also achieves a 20% margin over baselines. However, it is not sufficient to conclude that *SimGroupAttn* performs better overall due to the very limited number of images in complex classes. Further experiments are needed in this respect.

These results show potential benefits of incorporating population-level information in attention mechanism especially when the amount of intra and cross-image attention are optimized.

## 6 Conclusion

In this work, we introduced *SimGroupAttn*, a novel attention mechanism that enhances Vision Transformers by incorporating population context. Our approach enables image patches to attend to their most similar counterparts both within and across images in the same batch, effectively embedding population structure into masked image modeling. We evaluated *SimGroupAttn* on the PlantPathology dataset, comparing it against existing methods. The results show that our method improves the quality of reconstruction and enhances the performance of representation learning for complex classes. Furthermore, we observed gains in classification accuracy under certain configurations. We hope this framework will support and inspire future research in this direction.

## 7 Limitations & Future Work

While *SimGroupAttn* delivers clear gains over existing masked image modeling frameworks, it also comes with certain limitations. The cross-image attention mechanism introduces running time overheads. Appendix A.1 is the analysis of running complexity. Our experiments were also restricted to plant disease dataset therefore further evaluation on other domains is valuable to test broader applicability. Future work will focus on developing more efficient attention indexing to lower computational cost and possibly on adaptive strategies that balance intra- and cross-image attention.

# Acknowledgements

We would like to express our sincere gratitude to our colleagues at Crop Science, Bayer AG, and the University of Bonn for their invaluable support, guidance, and encouragement throughout the course of this work. Their insightful discussions, constructive feedback, and expertise significantly contributed to shaping the direction and outcomes of this study. We are particularly grateful for the collaborative environment fostered by our teams, which facilitated the exchange of ideas and problem-solving that was essential for the success of this project.

We also acknowledge the computational resources and technical support provided by Bayer AG and the University of Bonn. Access to these advanced resources was instrumental in enabling the extensive experiments and large-scale data analyses conducted in this study, without which achieving the reported results would not have been possible.

Finally, we extend our gratitude to the reviewers for their careful reading, thoughtful comments, and valuable suggestions. Their feedback greatly enhanced the clarity, rigor, and overall quality of this paper, and we deeply appreciate the time and effort they invested in providing such constructive guidance.

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

# A Running Complexity

Table A.1 presents the average inference time per image (s/image) across different Vision Transformer backbones. All experiments were conducted on a single NVIDIA A10G GPU (24 GB memory) using untrained models, which does not affect inference time. This choice allows us to evaluate much larger models than would otherwise be feasible and provides practical insights for use cases where training large models is beneficial. For each configuration, inference was performed on a single image 1000 times, and the mean runtime was reported.

As expected, inference time increases with model scale—from ViT-B to ViT-H—due to the added cost of cross-sample attention and similarity-matrix computation. **SimGroupAttn** introduces a moderate computational overhead relative to SimMIM, with the maximum slowdown ranging from $1.46\times$ for ViT-B to $1.63\times$ for ViT-H. This overhead scales consistently with model size, indicating predictable performance costs when adopting group-attention mechanisms in larger models.

From Table A.1, we observe that while the **SimGroupAttn** method is slower than SimMIM and MAE, the increase is modest compared to the benefits of enabling richer attention patterns. The results also show that the relative overhead does not grow disproportionately with model size, which is encouraging for practical deployment in large-scale settings. Using untrained models ensures that these measurements reflect the computational characteristics of the architecture itself, rather than training dynamics, making the analysis broadly applicable to scenarios where large models are used for inference or pretraining.

# B Reconstruction Quality Quantification

Table B.1 presents the per-class reconstruction performance measured by the $L_1$ loss, where lower values correspond to better reconstruction. All experiments were performed using a model pretrained on the PlantVillage dataset, without joint training on the PlantPathology dataset, with a batch size of 128.

Across the 12 categories, **SimGroupAttn** consistently matches or outperforms both MAE and SimMIM in most categories, demonstrating the benefit of incorporating group-attention mechanisms for image reconstruction. For example, in more challenging categories such as *powderymildew* and *powderymildewcomplex*, **SimGroupAttn** achieves substantial improvements over both baselines, reducing the $L_1$ loss by 18.2% and 13.4%, respectively. Similarly, notable gains are observed for *complex*, *rust*, *frogeyeleafspotcomplex*, and mixed-disease categories like *rustfrogeyeleafspot*, with improvements ranging from approximately 3.5% to 5.9%.

In some categories, such as *healthy* and *scab*, **SimGroupAttn** shows minor differences compared to SimMIM (0.6% worse and 5.6% worse, respectively). Overall, the results suggest that **SimGroupAttn** is particularly effective in capturing fine-grained and complex spatial patterns, leading to more accurate reconstructions in most cases. The negative $\Delta(\%)$ values in most categories highlight the robustness of the method, showing that the architectural improvements can be translated into measurable gains in reconstruction performance.

Importantly, these reconstruction improvements have practical implications for downstream tasks: by producing more precise representations during pretraining, **SimGroupAttn** can enhance performance on tasks such as classification, segmentation, or anomaly detection, especially in datasets with complex or overlapping patterns, especially with sample correlated features, as highlighted by the results in previous sections. This demonstrates the benefits of this method in applications where fine-grained and high-quality feature learning is critical.

| ViT | #Params (M) | MAE | SimMIM | SimGroupAttn (Ours) | Max Slowdown |
|-----|-------------|-----|--------|---------------------|--------------|
| ViT-B | 86 | 0.0149 | **0.0146** | 0.0213 | 1.46× |
| ViT-L | 307 | **0.0258** | 0.0291 | 0.0410 | 1.59× |
| ViT-H | 632 | **0.0335** | 0.0391 | 0.0545 | 1.63× |

**Table A.1.** Average inference time (s/image) for different ViTs using MAE, SimMIM and **SimGroupAttn**. The reported values represent the mean running time to reconstruct a single image, showing the computational overhead introduced by the group-attention mechanism relative to the SimMIM baseline. *Max Slowdown* represents the largest ratio between the running time of our method and the fastest method for each ViT.

| Category | MAE | SimMIM | SimGroupAttn | $\Delta(\%)$ |
|----------|-----|--------|--------------|--------------|
| complex | 0.0915 | 0.0700 | **0.0659** | -5.9 |
| healthy | 0.0828 | **0.0688** | 0.0692 | +0.6 |
| rust | 0.0699 | 0.0570 | **0.0538** | -5.6 |
| scab | 0.0941 | **0.0683** | 0.0721 | +5.6 |
| frogeyeleafspot | 0.0929 | 0.0734 | **0.0721** | -1.8 |
| frogeyeleafspotcomplex | 0.0850 | 0.0640 | **0.0609** | -4.8 |
| powderymildew | 0.0847 | 0.0777 | **0.0635** | -18.2 |
| powderymildewcomplex | 0.0788 | 0.0627 | **0.0543** | -13.4 |
| rustcomplex | 0.0802 | 0.0663 | **0.0640** | -3.5 |
| rustfrogeyeleafspot | 0.0758 | 0.0595 | **0.0568** | -4.5 |
| scabfrogeyeleafspot | 0.0959 | 0.0732 | **0.0725** | -1.0 |
| scabfrogeyeleafspotcomplex | 0.0932 | 0.0703 | **0.0672** | -4.4 |

**Table B.1.** Per-class reconstruction $L_1$ loss (lower is better). $\Delta(\%)$ represents the relative change of **Sim-GroupAttn** compared to min(MAE, SimMIM). Negative values indicate a lower loss, corresponding to higher reconstruction quality, while positive values represent an increase in loss, reflecting reduced reconstruction precision.

