# OpenReview forum: "SimGroupAttn: Similarity-Guided Group Attention for Vision Transformer to Incorporate Population Information in Plant Disease Detection"
_NLDL.org/2026/Conference — NLDL 2026 Spotlight_

### Official Review · Reviewer_cXzx · 2025-09-19
**While incorporating cross-image information in attention is a novel and appealing idea, the scope of applicability for SimGroupAttn remains unclear, and the current experimental evidence does not convincingly demonstrate its effectiveness.**

**Rating:** 2
**Confidence:** 4
**Final Rating:** 4
**Final Confidence:** 4

**Summary:**

This paper introduces a novel attention mechanism, similarity-guided group attention (SimGroupAttn), for image classification.
Unlike conventional attention, SimGroupAttn assigns weights not only within patches of a single input image but also across patches from other images in the same batch.
By leveraging information from multiple images, the method can capture subtle features, such as plant disease symptoms, within a representation learning framework like masked image modeling (MIM).
Experiments show that SimGroupAttn produces richer feature representations and achieves higher accuracy on the plant disease classification task.

**Strengths:**

- S1: Extending attention to gather information across images, rather than restricting it to a single image, is an interesting idea.
- S2: The paper is well-written and easy to follow.

**Weaknesses:**

- W1: It remains unclear what types of data benefit from SimGroupAttn. The concept of population-aware classification (L124) is not well-defined. In standard machine learning, samples are typically assumed to be i.i.d., meaning they do not share information. While it is reasonable that MIM reconstruction improves with SimGroupAttn, the mechanism why it contributes to downstream accuracy is not well explained, given the i.i.d. assumption. Clarifying the scenarios or assumptions under which cross-image patch sharing is effective would strengthen the generality of SimGroupAttn beyond plant disease classification.
- W2: The structure of the method and experiment sections could be improved. For instance, the model architecture description in Sec. 3.2 reads more like an experimental setting than part of the proposed method. Conversely, the training procedure described in the experiments (e.g., loss function, batch construction) should be described in the proposed method.
- W3: The inference procedure after training is unclear. Should inference also require batching of data? During training, batches are constructed in a non-standard way (L282), but in practice inference data may be sampled randomly. It is unclear whether this discrepancy introduces performance issues.
- W4: Some experimental results rely on subjective assessment:
  - The reconstruction results in Sec. 5.1 and Fig. 3 should be compared quantitatively (e.g., with MSE), as reconstruction quality is critical for demonstrating the effectiveness of SimGroupAttn.
  - Attention map visualizations would provide stronger evidence for the claim in L496–500 `These results suggest that incorporating population-level information helps the model better distinguish between groups, as cross-image attention encourages the formation of more informative latent representations.`
- W5: The computational overhead of SimGroupAttn compared to standard attention is not discussed.

**Final Justification:**

I appreciate the authors' rebuttal and further explanations. My concern about the types of data for which SimGroupAttn is effective and how it improves downstream accuracy has been addressed.

I would like to lean my rating toward accept as long as the quantitative evaluation (W4) is addressed in the revised paper.

**Justification:**

While incorporating cross-image information in attention is a novel and appealing idea, the scope of applicability for SimGroupAttn remains unclear, and the current experimental evidence does not convincingly demonstrate its effectiveness.

---

> ### Author Rebuttal · Authors · 2025-10-22
>
> Thank you very much for your insightful feedback and constructive suggestions, which provide valuable guidance for improving the clarity and overall quality of our work.
> Regarding your suggestion on section structure, we will revise and reorganize the manuscript in the final version.
> The concept of population-awareness refers to modeling shared structures or correlations among samples originating from the same population or group. In many biological datasets—such as those involving plants, cells, tissues, or animals—samples are not strictly independent and identically distributed (i.i.d.). Instead, they are drawn from populations with shared traits, so samples exhibit correlations and reflect population-level similarities. In population-aware classification, the model does not treat each sample as fully independent; rather, it leverages relationships among samples from the same underlying population.
> The improvement in downstream task accuracy arises from the internal clustering mechanism of SimGroupAttn. Specifically, the cross-image attention acts as an implicit clustering mechanism, encouraging correlated samples to be closer in representation through patch reconstruction using cross-sample patches. While standard attention mechanisms primarily model i.i.d. relationships, SimGroupAttn additionally captures inter-sample correlations. Consequently, it is particularly well-suited for applications in biological, agricultural, and medical domains, where samples often share correlations due to underlying biological, genetic, or environmental factors. We will include additional clarifications on this mechanism in the final version.
> In this work, inference data is batched in the same way as during training. We acknowledge that this is not ideal in practice, as cross-image attention requires more than one sample per batch. Our current focus is on demonstrating the novelty and potential of the methodology; therefore, we did not further investigate this aspect. Nonetheless, decoupling the model from specific batch construction is clearly a valuable direction for future work.
> Finally, we acknowledge that a more thorough analysis of computational cost would strengthen our work. We are actively optimizing runtime and addressing key bottlenecks, and we plan to provide a more detailed discussion along with quantitative results in the final version of the paper.

---

### Official Review · Reviewer_iKdi · 2025-10-08
**Good paper and method, but limited scope in evaluation**

**Rating:** 4
**Confidence:** 2

**Summary:**

The paper proposes SimGroupAttn, a new attention mechanism that allows Vision Transformer (ViT) models to attend not only within individual images but also across similar patches in other images within the same batch. This mechanism introduces population-level attention guided by learned cosine similarity scores. Standard ViTs and masked image modeling (MIM) frameworks such as MAE and SimMIM indeed restrict attention to intra-image tokens, limiting the ability to model shared visual patterns across samples. The authors define similarity-guided cross-image attention by constructing a sparse attention graph where each patch attends to its top-M intra-image and top-N cross-image neighbors.

**Strengths:**

SimGroupAttn’s key insight—leveraging population-level context for better feature disentanglement—is highly relevant for agricultural and biological imaging domains. In these settings, many subtle patterns (e.g., leaf discolorations, small lesions) are ambiguous when seen in isolation but become discriminative when compared across similar samples. This paper formalizes this intuition into a trainable attention module, allowing ViTs to encode sample-level relationships. This idea could have broader implications for other domains with inter-instance correlations (e.g., medical imaging, histopathology, or quality inspection). In linear probing, SimGroupAttn improves Top-1 accuracy by 6.5% on complex (multi-disease) classes. In classification, SimGroupAttn achieves up to 63.1% Top-1 accuracy, surpassing both SimMIM (56.9%) and MAE (59.6%) under equivalent conditions.

**Weaknesses:**

SimGroupAttn introduces significant computational cost due to its cross-image attention mechanism. The authors note that runtime complexity is roughly 3× higher than baselines when smaller patch sizes or larger batches are used. This scalability limitation raises concerns about practicality for large-scale datasets.

Because attention operates across batches, batch composition becomes a critical design factor. The method requires grouping images of the same species to avoid irrelevant cross-species attention. This introduces sensitivity to data sampling strategy.

All experiments are conducted on plant disease datasets. While appropriate for the problem setting, this narrow scope limits the claim of generality.

Although the ablation studies explore (M, N, B), the model’s optimal configuration (M=30, N=19, B=32) seems dataset-specific. Performance degrades when N is too high or too low. This suggests sensitivity to the ratio of intra- vs. cross-image patches.

For single-disease (“simple”) classes, SimGroupAttn yields almost no improvement (≤0.5% over baselines). Thus, its benefits appear limited to complex datasets with high inter-sample redundancy. While this supports the authors’ motivation, it also indicates that the method may not generalize to simpler or more linearly separable domains.

**Justification:**

Integrating population-level attention into ViTs represents a promising and underexplored direction for self-supervised vision models. The method is innovative, well-engineered, and empirically validated.

---

> ### Author Rebuttal · Authors · 2025-10-22
>
> We sincerely appreciate your insightful comments and constructive suggestions, which have been invaluable in guiding improvements to our work.
> First, we would like to clarify a factual point regarding your last observation: single-disease classes do show improvements over the baselines under the joint-training setting, as shown in Table 5, although they do not outperform the baselines in the linear probing task, where the performance of complex classes dominates. For instance, the best performance of SimGroupAttn exceeds that of the baseline by up to 8.6% for the ‘Healthy’ class and 6.5% for the ‘Scab’ class (both over MAE), demonstrating its potential advantage in linearly separable domains.
> We acknowledge that runtime is an operational bottleneck. We have already identified several components for optimization to reduce computational cost and will include more detailed discussion in the final version. Regarding batch composition, it is indeed true that our method’s cross-batch attention makes batch construction an important factor. In the current work, our primary focus has been on demonstrating the potential of this novel approach; therefore, we did not explore this aspect in depth. Nonetheless, refining the batching strategy to reduce sensitivity to data sampling and enhance compatibility with broader deployment scenarios represents a valuable direction for future work.

---

### Official Review · Reviewer_4fj5 · 2025-10-13

**Rating:** 4
**Confidence:** 4

**Summary:**

This paper proposes SimGroupAttn, a novel attention mechanism for Vision Transformers (ViTs) that extends self-attention beyond intra-image scope to incorporate cross-image similarity. The method allows patches from one image to attend to similar patches in other images within the same batch, thereby embedding population-level information. It is applied to plant disease detection, where such cross-sample relationships can improve recognition of subtle or overlapping symptoms.
Experiments on PlantVillage and PlantPathology datasets demonstrate that SimGroupAttn outperforms SimMIM and MAE, achieving up to 6.5% improvement in linear probing and 3.5% improvement in classification accuracy. The approach shows especially strong performance on complex classes involving multi-disease infections. The paper also includes ablation studies on intra- vs. cross-image attention balance (M, N) and batch size.

**Strengths:**

Novelty: The idea of incorporating population-level attention into ViTs is innovative and addresses an important limitation of standard self-attention (lack of cross-sample reasoning).

Soundness: The proposed mechanism is mathematically well-formulated and clearly integrated into the ViT framework. Equations (1–3) are correct and consistent with standard attention formulation.

Empirical Validation: The experiments are extensive, covering reconstruction, linear probing, classification, and ablation analyses. The results consistently favor SimGroupAttn, particularly in complex disease cases.

Clarity: The paper is well-organized, with detailed methodology and clear visualizations (e.g., Figures 1–4).

Significance: Population-aware modeling could generalize beyond agriculture to medical imaging or other biological domains, representing a meaningful conceptual advance.

**Weaknesses:**

Limited Scope of Evaluation: Experiments are restricted to plant disease datasets; validation on a more diverse benchmark (e.g., ImageNet, CIFAR, or medical imaging) would strengthen claims of general applicability.

Computational Cost: The paper admits that cross-image attention increases training cost by ~3×, which may limit scalability for large datasets. No runtime or memory usage metrics are reported.

Statistical Robustness: Improvements (e.g., +3.5% classification gain) are modest and not accompanied by significance testing. Some complex-class results rely on very small sample sizes.

Batch Construction Sensitivity: The method’s dependence on batch composition (grouping by species) may hinder generalization and complicate practical deployment.

Clarity on Similarity Learning: It remains unclear how the cosine similarity module is optimized jointly with ViT weights—more implementation details (loss contribution, gradient flow) would be helpful.

**Justification:**

This paper presents a well-motivated and technically sound improvement to ViTs by extending attention to cross-sample relationships through similarity-guided grouping. The approach is original, clearly described, and supported by consistent empirical results. Despite limited dataset diversity and higher computational cost, the method’s conceptual contribution—embedding population-level context into self-attention—is significant and broadly relevant. Overall, the strengths outweigh the weaknesses, warranting an accept recommendation.

---

> ### Author Rebuttal · Authors · 2025-10-22
>
> Thank you very much for your thoughtful and insightful feedback and constructive suggestions, which have provided valuable guidance for improving our work.
> We acknowledge that a more detailed analysis of computational cost would further strengthen our work. We are actively working on optimizing the runtime and addressing computational bottlenecks, and we plan to include a more comprehensive discussion and quantitative results in the final version of the paper.
> Regarding the limitation of the evaluation scope, one concern about many standard benchmarks (e.g., ImageNet, CIFAR) is that they usually do not exhibit the inter-sample population dependencies characteristic of plant disease data—an aspect central to the problem our method addresses. For this reason, we went for this plant disease dataset instead of the common benchmark datasets.
> With respect to batch construction, there is indeed room to improve the method’s compatibility with more general and practical deployment settings. Although our current work focuses primarily on demonstrating the potential of this methodology, refining the batching strategy to enhance scalability and ease of use in broader contexts will be a very beneficial improvement.
> Finally, regarding the cosine similarity module, it is optimized implicitly during training. Specifically, cosine similarity is employed to identify the most relevant patches for reconstructing a masked patch, thereby influencing the loss trajectory through guided patch selection rather than contributing directly to the loss itself. We can include additional implementation details in the final manuscript to clarify this mechanism more explicitly.

---

### Meta-Review · Area_Chair_AhBU · 2025-10-29

**Recommendation:** Accept (Poster)
**Confidence:** 4

**Metareview:**

This paper appears to be innovative, well-validated, and clearly written. Moreover, it seems that most, if not all, of the reviewers’ concerns have been adequately addressed and can be incorporated into the camera-ready version.

---

### Decision · Program_Chairs · 2025-11-05

**Decision:**

Accept (Spotlight)

**Comment:**

We recommend an oral and a poster presentation given the AC and reviewers recommendations.

A spotlight presentation refers to a poster selected for an oral highlight but not designated as a full oral presentation per the AC’s recommendation.